# Traffic jam by GPS: A systematic analysis of the negative social externalities of large-scale navigation technologies

Eve Schade[1]*, Gian-Luca Savino[1], Yasemin Gunal[2], Johannes Schöning[1]

1 University of St. Gallen, St. Gallen, Switzerland, 2 University of Michigan, Ann Arbor, MI, United States of America

* eve.schade@unisg.ch

**Data Availability Statement:** The underlying corpus of our study is based on articles obtained from third-party sources. While we are unable to provide the full articles due to copyright

## Abstract

The increased usage of navigation technologies has caused conflicts in local traffic management, resulting in congested residential areas among other challenges for residents. This paper uses content analysis to investigate such negative social externalities within local communities and neighbourhoods. Through a corpus of 90 news articles about traffic incidents caused by navigation technologies, we identified negative traffic and safety-related externalities, including congestion, damage, pollution, and accidents. We also report on countermeasures by local communities and governments, including street closures, speed limit reduction, and turn bans. Based on our results, we discuss the implications for designing mobile navigation technologies that reduce negative social externalities.

## Introduction

In-car navigation systems introduced many individuals to the convenience and practicality of mobile navigation technologies [1, 2]. Today, mobile devices, empowered through advanced positioning capabilities, such as GPS, Wi-Fi and cellular connection, have become the de-facto standard. Mobile navigation applications running on those mobile devices have evolved from a novelty to a near-necessity for navigating both familiar and unfamiliar environments [3–6]. Initially, these technologies employed shortest-path algorithms to compute an efficient route from point A to point B [7]. However, as they advanced and more criteria, such as live traffic data [8] and user preferences [9–11], were incorporated, the routing algorithms became significantly more advanced [12]. Today, many mobile navigation technologies leverage real-time data to provide dynamic routing suggestions that adapt to changing road and traffic conditions [8]. The set of criteria used for further route optimisation keeps expanding. Through research, preferences for different routing criteria are understood and developed into alternative routing approaches that focus on safety [13], pleasantness [14], and scenic value [15]. As this development continues and routing becomes more optimised for individual users' preferences and use cases, it will inadvertently impact the overall traffic [16, 17]. For instance, if routes are designed to avoid urban areas with high crime rates, as in safe routing approaches [13], it can lead to unintended negative social externalities (NSEs) [17]. NSEs in the context of large-scale

restrictions, we have included a comprehensive dataset in the supplementary materials of our manuscript. This dataset contains detailed references for each article, including titles, descriptions, and search terms used. This enables interested researchers to identify and access the original articles from the respective publishers, should they still be available.

**Funding:** Funded by: Swiss National Science Foundation (SNSF) Grant number: 207430 Recipients: Prof. Dr. Johannes Schöning URL: https://data.snf.ch/grants/grant/207430 Our project's funders had no role in study design, data collection, analysis, decision to publish, or manuscript preparation.

**Competing interests:** The authors have declared that no competing interests exist.

routing are side effects resulting from a change in traffic routing based on personal or algorithmic decisions [17]. In the case of safety routing, one such externality would be the traffic reduction in certain "unsafe" urban areas, subsequently lowering their economic standing [17, 18].

Through the widespread use of navigation technologies and also alternative routing criteria, we will see increased unintended NSEs for residential areas. Thus, NSE will reduce the perceived liveability in such areas [19] as high motor vehicle traffic is often seen as a major threat to the quality of life in residential areas [20]. In addition, factors such as walkability of an area increases the liveability of a residential disctrict. Especially for younger people, heavy traffic and car parking decreased positive perceptions of the safety, friendliness, appearance, play facilities and helpfulness of the people in their local area [21]. Kingham et al. [22] found that closing residential streets to through-traffic results in increased happiness and quality of neighbourhood relations for the residents. The impact and significance of NSE are not only discussed in various scientific communities but are also part of a public discourse. Various newspaper articles are discussing NSEs [23–27], the issues with navigation technologies are highlighted in artists' performances [28] or prominent TV shows like The Simpsons [29].

Therefore, this paper explores and systematically analyses the NSEs and captures the solutions currently employed to mitigate them. From a corpus of 3052 news articles, we identified and analysed 90 unique articles that reported traffic problems in residential areas caused by mobile navigation technologies. We analysed the articles using content analysis [30, 31], identifying key themes and categories related to the NSEs caused by mobile navigation technologies and the potential solutions proposed by various stakeholders, such as residents and local governments. We use these insights to discuss how these negative externalities can be mitigated through design implications for mobile navigation technologies. Therefore, in this paper, we make the following contributions:

1. We perform the first systematic categorisation of NSEs induced by the large-scale use of mobile navigation technologies.

2. We categorise existing solutions that are used to mitigate these NSEs.

3. We propose design implications for navigation technologies to mitigate the NSEs caused by them.

Our analysis revealed eight NSEs of large-scale routing by navigation systems: increased congestion, influx of heavy goods vehicles in narrow streets, breaking of traffic laws, inconveniences for communities, concerns regarding safety risk, damage to surfaces, pollution, and accidents. Many residents reported the "feeling of being trapped" in their neighbourhoods, unable to navigate their streets safely due to the influx of non-local traffic. The non-local traffic leads to gridlock and extended waiting times at intersections. In attempts to bypass congested areas or secure shortcuts, drivers engaged in speeding and reckless driving behaviour, creating additional safety hazards. The increased traffic volume and speeding, in particular, posed significant risks to pedestrians and cyclists navigating these residential areas. Furthermore, we categorise existing solutions implemented to tackle these problems, including additional signage, implementing low-traffic neighbourhoods (LTN), and collaborating with mobile map application providers to share traffic information. We find that most local government solutions have a high success rate. Residents are also experimenting with solutions to mitigate the local externalities, resulting in much lower success rates. The solutions we identify address many of the discussed externalities directly. However, we propose to generate solutions that seek to mitigate the externalities and prevent them entirely. To approach such sustainable solutions to social externalities caused by the large-scale use of mobile navigation technologies, we

propose informing the users of mobile navigation technologies better about the consequences of their navigation behaviour. We discuss design implications for mobile navigation applications, such as highlighting the potential impact of navigation decisions on residential areas and using such information directly in routing algorithms to avoid creating these problems.

Our findings contribute to the growing body of research on the impact of technology on society [32, 33] and highlight the need for a collaborative approach between governments, residents, city planners and technologists to address the unintended consequences of navigation technologies on residential areas [34].

## Methods

NSEs of navigation technologies on traffic within local communities and neighbourhoods have emerged as a novel phenomenon during the past decade. Even though there is widespread news coverage on those externalities and scientific interest in the topic [23], no data set exists that allows for a systematic analysis.

This section describes our method to develop our corpus of relevant news articles. We based our approach and method on the work of Lin et al. [35], who analysed a similar phenomenon (Death by GPS) in earlier work. We also adapt Stryker et al.'s [30] method to highlight the validity of the dataset by employing a two-phase process: (1) developing a corpus of relevant news articles, followed by (2) an expert-led coding analysis.

### Phase 1: Corpus development

To develop our corpus of international articles about the side effects of navigation technologies, we used the NexisUni platform provided by LexisNexis [36]. To identify relevant articles, we generated a set of English keywords that were succinctly used to create three search terms. Our data set included articles published between 2010 and 2023. Fig 1 illustrates the development of our corpus, which is further described in the following paragraphs.

Following best practices for keyword generation [30], we created meaningful keywords from related work in the first step (a). This initial set was extended through synonym generation. Based on the resulting keywords, two authors of the paper, also domain experts, generated the total set of 92 keywords by reviewing an initial set of articles from the NexisUni platform to determine the final set of keywords. This process resulted in extracting contextually appropriate keywords and identifying commonly recurring synonyms, such as *rat-running* for *selfish use* of navigation technologies. Additionally, we added 20 keywords that can be used to exclude specific results from the search as they were mostly irrelevant and resulted in misleading search results (e.g. *aircraft, trial, patent*). Refer to S2 File for the complete set of included and excluded keywords.

Based on these keywords, we derived three search terms (b) iteratively over several expert-led iterations. These were individually developed and optimised for precision. Each search term generation involved analyses by two authors who went through three iterative rounds for each search term in accordance with the method by Stryker et al. [30]. In each round, the first 100 articles that resulted from the search term were screened, and the precision of the search term was calculated using the ratio of relevant and irrelevant articles. The search term was adapted if necessary, and a second round of screening 100 articles was performed. For each search term, three rounds were performed. After three rounds, the search term with the highest precision was selected. The first 1000 articles resulting from each of the three search terms were reviewed during the final screening process. Among these, the primary search term, which integrated the most keywords from the set, achieved a precision of 0.7%, resulting in 70 relevant articles (see S3 File for the full list of articles). The secondary term achieved a precision

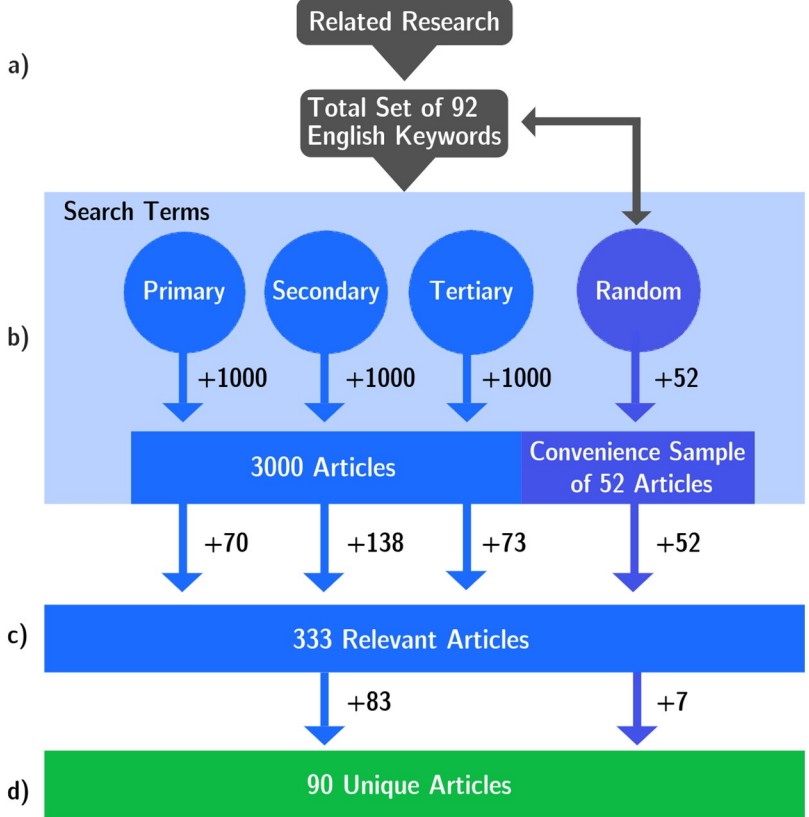

**Fig 1. Process of corpus development.** (a) The keywords were derived from relevant literature and various random search terms. (b) The article search was conducted using three developed search terms drawn from the comprehensive keyword set, resulting in 3000 articles and a convenience sample resulting from random search terms, yielding 52 articles. Refer to S1 File for the Random Sample. (c) All 3052 were checked for relevance in the next step, resulting in 333 relevant articles. After removing duplicates, opinion articles, and summary articles (d), 90 were identified as unique and relevant, constituting the final corpus.

of 1.38%, resulting in 138 relevant articles (see S4 File for the full list of articles). The tertiary search term achieved a precision of 0.73%, resulting in 73 relevant articles (see S5 File for the full list of articles).

In addition to the 3000 articles screened using the three crafted search terms, we performed unstructured searches on NexisUni to uncover articles missed by the three search term. This allowed us to build up a convenience sample of additional articles (see S1 File for the full list of articles). In the end, our corpus comprised 3052 articles, which were gathered using three specific search terms, resulting in 1000 articles each plus random sampling, which resulted in 52 additional articles. In an intermediate step (c), two authors reviewed all 3052 articles for relevance. This was primarily done by reading the article title. If the title did not clearly indicate whether the article was relevant, the authors read it to assess its relevance. This screening method resulted in 333 (70 + 138 + 73 + 52) articles.

In the next step (d), we removed any duplicates that appeared in our set of articles. This included cases where the same incident was reported by more than one news outlet. We also excluded articles that only provided high-level discussions on several scenarios or were opinion columns and thus did not describe a specific incident. After the final screening process, our corpus yielded 90 relevant articles, with 7 articles coming from our convenience sample.

Each involves a unique story about social externalities caused by mobile navigation technologies.

## Phase 2: Expert-led coding & analysis

During the second phase, two coders analysed and filtered the 90 collected news articles [35]. Both coders have a scientific background in geographic information sciences and HCI. The coding process involved the development of a codebook, concentrating on essential categories and establishing corresponding subcategories for comprehensive evaluation. Two coders coded a representative sample of 10% of the data using an open-book coding approach in line with Blandford et al. [37]. Through iterative discussions, an initial codebook was established, which was then used by one single author to code the remaining material. Any newly added codes were discussed in the process between both authors. After the material was fully coded, code occurrences were used to aggregate similar news stories into groups and to identify overarching themes across the different externalities, according to Blandford et al. [37]. These themes and groups are used to structure the findings in the Results section. Refer to S6 File for the coded articles of the corpus.

## Results

In this section, we describe the main findings from our analysis. Through the coding process, we identified two main themes: We describe (1) the different NSEs that occur due to the large-scale use of navigation technologies and present (2) the different solutions that residents and governments proposed to counteract the negative externalities. All results are reported in whole numbers and percentages. For percentages, the amount is always reported in relation to the total number of 90 articles. Articles could be, and often were, discussing both externalities and solutions. In this regard, 69 articles discussed traffic-related and 35 safety-related externalities.

### News article dates & origins

The final corpus consists of 90 articles and comprised of news stories from 6 mostly English-speaking geographical regions and countries: UK (50, 55%), USA (29, 32%), Australia (6, 7%), Canada (2, 2%), India (2, 2%), and New Zealand (1, 1%). Stories came from 64 districts, cities, and villages within these regions. Based on NexisUni data, these articles were published between 2010 and 2023 with the following number of publications per year as shown in Fig 2: 2010 (1, 1%), 2011 (7, 8%), 2013 (1, 1%), 2014 (6, 7%), 2015 (4, 4%), 2016 (6, 7%), 2017 (13, 14%), 2018 (14, 16%), 2019 (12, 13%), 2020 (7, 8%), 2021 (9, 10%), 2022 (9, 10%), 2023 (1, 1%).

### Negative Social Externalities (NSEs)

During the coding process, we identified two distinct categories of externalities in urban traffic that have arisen due to navigation technology usage. These include traffic-related externalities (69, 77%) and safety-related externalities (35, 39%). Table 1 gives an overview of all externalities. Traffic-related externalities are concerned with changes that directly affect road conditions or road users in traffic situations. In contrast, safety-related externalities are concerned with changes that affect residents, pedestrians, and other road users not directly involved in active traffic. Articles could be counted towards both categories if they discussed multiple externalities. In the following sections, we examine these two subsets of issues, as well as their negative impact on society and the various government and citizen-implemented solutions designed to counteract their effects.

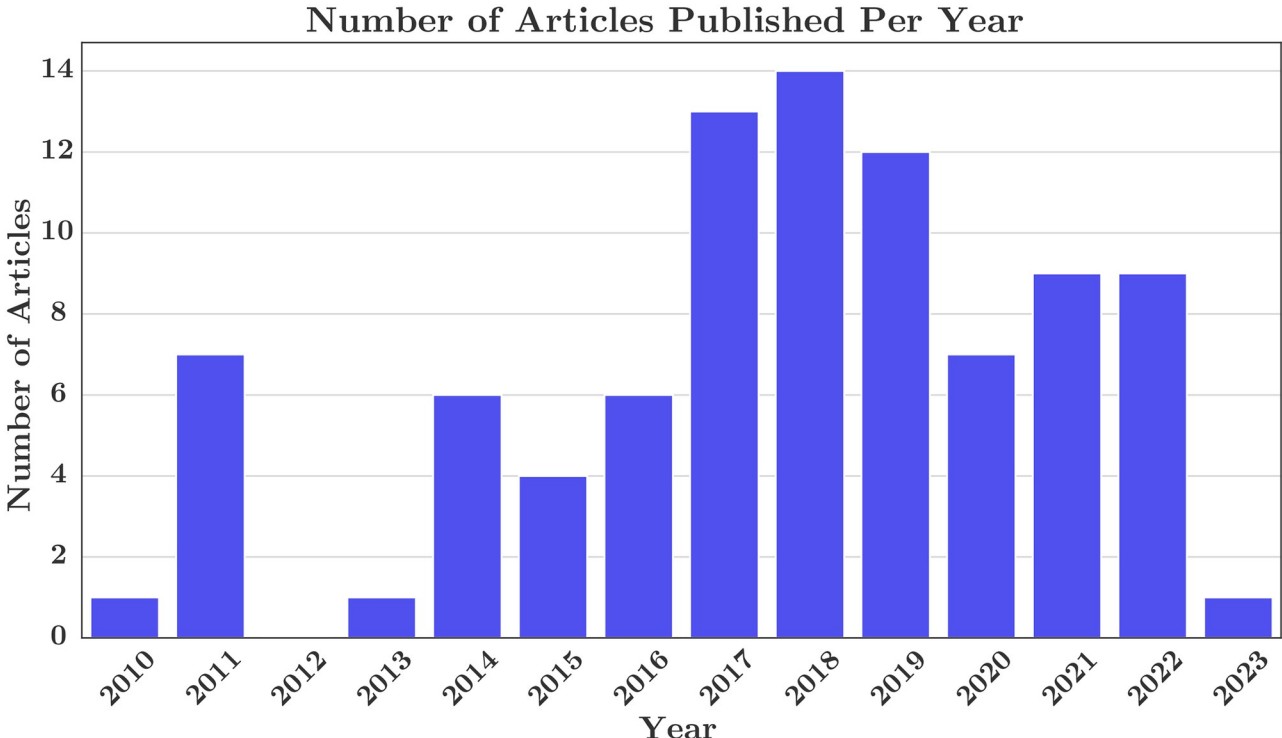

**Fig 2. News articles published per year.**

**Traffic-related externalities.** Externalities categorised as traffic-related involve one of four topics of general traffic issues due to navigation technology usage. These include an excess of vehicles contributing to congestion (34, 38%), a significant presence of heavy goods vehicles (HGVs) involved in traffic disruptions and accidents (21, 23%), violation of traffic laws (16, 18%), along with daily inconveniences or interferences imposed on surrounding communities (13, 14%).

Of the articles discussing traffic-related externalities, 34 (49%) indicated congestion as a primary traffic concern and addressed issues such as traffic jams, clogged streets, and rush hour. This congestion is often caused by navigation technology through redirection of traffic from highways or larger streets onto smaller side streets and into quiet neighbourhoods. In the

**Table 1. Distribution of NSEs.**

| Externality | Articles | # (%) |
|---|---|---|
| *Traffic-Related Externalities:* | | |
| Congestion | [24, 27, 38–69] | 34 (49%) |
| Heavy Goods Vehicles | [49, 50, 52, 54, 57, 70–85] | 21 (30%) |
| Traffic Laws | [40, 63, 75, 81–83, 86–95] | 16 (23%) |
| Inconveniences for Communities | [24, 38, 46, 94, 96–104] | 13 (19%) |
| *Safety-Related Externalities:* | | |
| Concerns Regarding Safety Risks | [24, 50, 77, 87, 93, 100, 105–112] | 14 (40%) |
| Damage to Surfaces | [51, 52, 70, 79, 80, 85, 90, 113–115] | 11 (31%) |
| Pollution | [51, 52, 86, 93, 94, 105, 116, 117] | 8 (23%) |
| Accidents | [49, 51, 58, 60, 65, 98, 113, 118, 119] | 9 (26%) |

corpus, 59 (66%) articles raise the issue of redirection, where navigation technology displaces traffic by suggesting residential routes or shortcuts that become overcrowded. This means the congestion is transferred from major highways to smaller side streets, as discussed by Associated Press International in 2014 when in Sherman Oaks, Los Angeles, traffic increased substantially on the residential streets near the "busiest urban freeway in America" as drivers began using Waze to avoid rush-hour congestion [110].

HGVs were the second most frequently discussed topic within the articles and were mentioned by 21 (30%). These articles focused on HGVs taking unsuitable roads, given their dimensions and weight, thus causing increased traffic or road blockages. For instance, in an article published by Kent Messenger in 2022, an HGV got stuck on train tracks when crossing through East Farleigh, a rural village in the UK with roads designed for smaller vehicles, on a route suggested by a navigation system and was forced to turn around to find an alternative route [85].

Furthermore, 16 (23%) articles in the traffic-related externalities category discussed how the usage of navigation technologies has contributed to the failure of drivers to adhere to traffic laws. Specifically, speeding out of frustration or to avoid waiting for pedestrians and illegal parking. This was reported in an example from The Telegraph, published in 2021, where a driver in Manchester City, using a *rat run*, was speeding, lost control, and hit a parked car [120]. In another example, The Toronto Star published an article in 2023 that discussed a small, highly congested street in Toronto, where drivers frequently make an illegal left turn onto a one-way street as a result of following their navigation system to reach the Allen Expressway [24]. Not only are these infractions of traffic laws, but they lead to hazardous situations for both drivers and pedestrians.

Lastly, inconveniences were discussed in 13 (19%) of the articles dealing with traffic-related externalities. This category included instances of individuals and communities expressing daily inconveniences due to traffic—for instance, scenarios like homeowners getting stuck in driveways and prolonged commute times. In an example discussed by The New York Times in 2017, in New Jersey, a college student was late to their class because their car was left parked on the street in front of their home and was "hemmed in by traffic" which had built up in their neighbourhood as a result of navigation technologies leading drivers to this street as a shortcut [26].

**Safety-related externalities.** In contrast to the traffic-related externalities, 35 articles reported safety-related externalities. These involved one of four categories that imposed a risk to the health and safety of humans or objects. These categories are concerns regarding safety risks (14, 16%), damage to properties and surfaces (11, 12%), noise and air pollution (8, 9%), and vehicle or pedestrian accidents (9, 10%).

Of the 35 articles raising concerns about safety-related externalities, 14 (40%) fall into the category of safety risks, such as unsafe walking conditions for pedestrians or dangerous driving environments for road users. Specifically, an article published by The Guardian in 2022 discussed that due to increased navigation technology usage in London, "pedestrians are 17% more likely to be killed or seriously injured on minor roads for every mile a vehicle travels than on major roads" [121].

Furthermore, 11 (31%) articles discussed damages caused by navigation technology. This has taken the form of road wear, vehicle wear, and damaged properties, each of which has yielded unintended consequences on urban traffic, particularly worsening the effects of congestion and pollution. Specifically, The Guardian wrote that in the UK in 2019, "brake wear, tyre wear, and road surface wear contributed to more than half of the particle pollution from road transport" [27], the effects of which are detailed in the next section. Additionally, damage to such surfaces often requires construction to resolve it. In 2016, The Washington Post

reported that in California, Maryland, and Oregon, construction-triggered reroutes by Waze turned informal shortcuts, which "used to spread by word of mouth" into "permanent routes", further exacerbating the traffic redirection issues [69].

Next, 8 (23%) articles discussed noise and air pollution as safety-related externalities caused by navigation technology usage. Not only does pollution occur as a direct result of damage to roads and tyres, but when applications divert traffic to other areas, it increases the pollution in those regions that experience increases in congestion-free of those symptoms. This is a rising concern of people living in residential areas experiencing this negative externality. Specifically, The Guardian published an article in 2020 highlighting a UK resident's concern: "We should not be sending more traffic down residential roads with high pollution levels" [27].

Lastly, 9 (26%) of the articles discuss accidents due to navigation devices. This externality can also be directly attributed to the issue of redirection due to navigation technology usage because drivers are often led to shortcuts or traffic-free roads that are far more difficult to navigate. For example, in 2017, The Evening Standard (London) discussed a situation where a driver navigated to a residential street that was commonly used as a rat run but "flipped up onto a parked car after the driver took a corner too fast" [60]. Such instances are indicative of how these navigation technologies cause an increase in accidents in urban traffic.

Our findings highlight the negative externalities, both traffic and safety-related, caused by the involvement of mobile navigation technologies in urban traffic. While most articles strongly emphasise the problems at hand, a few also mention solutions ideated and implemented by residents or the government to counter these externalities. The following section presents these solutions in further detail.

## Existing solutions against NSEs

We examined the successful and unsuccessful solution implementations highlighted by the articles in the corpus. Overall, in the 90 articles in the corpus, 34 distinct solutions were described (see Table 2 for a subsample). Of these, the most commonly attempted solutions included: closing streets to traffic (10, 11%), implementing signage (9, 10%), reducing speed limits (7, 8%), banning turns (6, 7%), and removing streets from the app altogether (5, 6%).

However, our analysis shows a difference in the type of solution offered when articles discuss traffic-related externalities versus safety-related externalities. The most frequent solutions for externalities categorised as traffic-related (69 articles) included closing streets to traffic (8, 12%), implementing signage (6, 9%), reducing speed limits (4, 6%)), implementing low traffic neighbourhoods (3, 4%), removing streets from the apps (3, 4%), banning turns (3, 4%), installing speed bumps (3, 4%), creating specialised navigation technology for HGVs (3, 4%), introducing one-way streets (2, 3%) and reporting fake accidents (2, 3%). For externalities categorised as safety-related (35 articles), the most frequently discussed solutions included

**Table 2. Distribution of existing solutions.**

| Existing Solutions | Articles | # (%) |
|---|---|---|
| Closing Streets | [38, 42, 48, 49, 53, 55, 67, 75, 78, 122] | 10 (11%) |
| Implementing Signage | [54, 57, 73, 74, 79, 90, 112, 114, 115] | 9 (10%) |
| Speed Limit Reduction | [43, 77, 89, 93, 94, 106, 108] | 7 8%) |
| Banning Turns | [42, 43, 46, 64, 98, 99] | 6 (7%) |
| Removing Streets from the Apps | [38, 61, 62, 98, 118] | 5 (6%) |
| Specialized Navigation Technology for HGVs | [71, 79, 85, 101] | 4 (4%) |
| Introducing Low-Traffic Neighborhoods | [68, 86, 96, 123] | 4 (4%) |

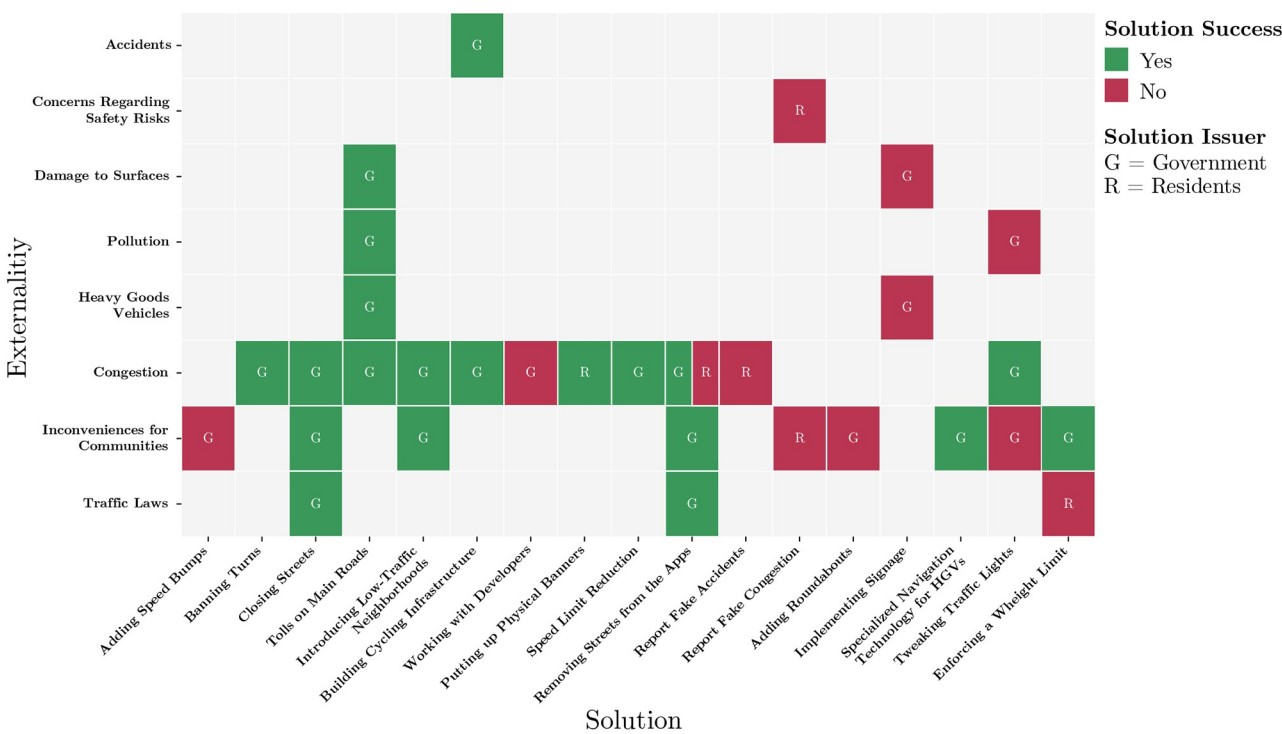

**Fig 3. Successfully (green) and unsuccessfully (red) applied solutions to different externalities by governments (G) and residents (R).**

reducing speed limits (5, 14%), implementing signage (5, 14%), removing streets from the apps (2, 6%) and reporting fake congestion (2, 6%).

Additionally, our findings indicate that the success of these solutions, when implemented, is highly dependent on the implementer. Compared to the government, residents have fewer options to influence the traffic in their neighbourhoods. Examples in our dataset range from putting up self-made signage, over trying to remove streets from navigation services to reporting fake accidents. We identified 32 externality-solution pairs across all articles. Fig 3 shows which solutions were applied to which externalities, by whom, and if they were successful. When comparing citizen and government-implemented solutions, government-implemented solutions have proven to be more successful. Of 26 government-implemented and tested solutions, 19 (59%) were successful, whereas out of the 6 resident-implemented and tested solutions, only 1 (16%) reported success. Even when both bodies attempted to implement the same solution, governments had higher efficiency. In this instance, the government-based removal of streets from a mobile navigation application was successful, whereas residents were not. When comparing residents and government-implemented solutions between traffic-related externalities and safety-related externalities, we find that a much larger number of the traffic-related solutions proposed by governments (21) compared to the ones proposed by residents (5). The same can be found for safety-related solutions, with 5 solutions proposed by governments and only 1 by residents. This can be attributed to the higher visibility and impact of government-implemented solutions. Solutions that necessitate alterations to existing regulations or laws—like prohibiting certain turns, introducing weight restrictions, or imposing additional charges for drivers using main roads—emerge as more feasible options for government implementation. This finding indicates that residents should consider encouraging high-visibility government action rather than trying to implement solutions themselves.

## Discussion

Mobile navigation applications such as Google Maps and Waze often dynamically re-route users through various lesser-known environments, causing multiple social externalities. As shown in simulations, e.g. by Thai et al. [16], while increased routing improves traffic efficiency, it also increases the average travel time for all. This dilemma then results in increased traffic in quiet neighbourhoods and side streets. Existing research corroborates these phenomena, indicating that they affect a city's overall traffic volume and the quality of life for individual residents and entire community neighbourhoods [17, 19–22]. With the first corpus of news articles describing these externalities and solutions local communities and governments try to counter them, our study further substantiates these findings.

Mobile map application providers, such as Waze, have already opened their systems to allow local governments to interact with their traffic and routing data. However, it is still challenging to make reasonable decisions, even if such data is available. One reason is that mobile traffic data can still be prone to errors, as illustrated by the fact that, e.g. traffic jams can easily be generated using hundreds of mobile devices in a hand wagon, as artists have shown in the past [28]. This paper takes a first step in making developers of mobile navigation technologies aware of the externalities described in this paper and the consequences their decisions have on other traffic participants and local communities. In our work, we provide an understanding of these externalities, their causes and potential solutions.

Generally, many articles revealed that the residents care a lot about the traffic within their neighbourhoods, and with good reason: NSEs reduce the quality of life in residential areas [19–22]. While not always rational, they experiment with reasonable solutions to overcome the social externalities. While on their own, residents are not very successful in implementing these solutions, the path through governments and legislation is often successful, as shown by our data. However, we argue that these solutions are just fighting the symptoms of large-scale use of navigation technologies.

To prevent social externalities, we can inform users about the consequences. Therefore, from a user perspective, we propose to augment existing navigation technologies with explainability [124]—providing the user with insights into why a particular route is suggested. For instance, users could be informed about potential consequences of their route, such as disrupting school dismissal times or causing undue noise during late-night hours. This context could allow drivers to make more informed decisions, and they might willingly choose to stay on a congested highway for a few extra minutes to minimise negative effects on local communities. Moreover, these broader criteria can be formally integrated into the design of routing algorithms. Traffic noise levels, $CO_2$ emissions, speed limits, street-specific volume caps, and randomisation [125] can be included in route optimisation, leading to more social route recommendations. This approach can also lead to more equitable distribution of traffic, reduced neighbourhood disturbances, and overall better alignment with the goals of sustainable urban living [126, 127].

We believe the shift from user-focused routing to a more community-centric and environmentally sensitive social routing approach could significantly mitigate the negative impacts of navigation technologies. However, further research must assess these adjustments' practical feasibility and potential implications.

### Limitations

We had to make several decisions during the corpus generation, which we want to highlight in this limitations section. During the last stage of the corpus generation, many articles had to be filtered out which reported stories about rat-running and congestion caused by

through-traffic or HGVs, but did not mention the use of navigation technologies as a cause of these problems. While navigation technologies could be a valid cause for many stories, we excluded these articles from the dataset as the use of navigation technologies was not explicitly reported.

Furthermore, Leonia in New Jersey (USA) is prominent and often mentioned in newspaper articles. In the debate on the effects of mobile navigation apps on residential neighbourhoods, Leonia is a kind of a poster child city regarding the debate on the effects of mobile navigation. It is, therefore, a good reference when discussing social externalities caused by navigation technologies. At the same time, the amount of news reporting on this one city might bias the data set. While we collected articles from diverse locations, we found a slight bias towards some geographic regions and more prominent news outlets, affecting the contents of the corpus (A snowball effect between the articles that reference each other or refer to similar events). Lastly, some articles differ in their current online version, cited in this paper, from the one from NexisUni in our corpus. We ensured that the NSE reported in the online article and the one in our corpus were identical.

## Conclusion

In this paper, we make three contributions to the field of navigation technologies. (1) We present the first systematic categorisation of NSEs induced by these technologies, shedding light on previously unexplored dimensions of their impact. (2) We offer comprehensive categorisations of existing solutions to mitigate these negative externalities, providing a valuable resource for researchers and practitioners seeking to address these pressing issues. Lastly, (3) we propose improvements for navigation technologies, underlining the importance of designing user-centred solutions and next-generation social routing techniques to mitigate the NSEs effectively.

NSEs have additionally permeated cultural resonance, thereby demonstrating the social significance of the topic. In addition to the referenced newspaper articles, concerns regarding navigation technologies are also underscored in artistic performances [28] and TV programmes such as "The Simpsons" [29]. In the recent episode of "The Simpsons" Lisa Simpson says, "Why did all the traffic suddenly appear on our street?". The app re-routes traffic to a quiet residential neighbourhood, turning it into a "gridlocked nightmare" [29]. Interestingly, The Simpsons applied one of the solution strategies identified in our corpus. They removed their street from the data set of a mobile navigation application but, in contrast to our corpus, were successful with it. Similarly, Weckert, a Berlin-based artist, also exploited the effects by using 99 mobile devices to deceive Google Maps into heavy traffic on the streets he walks [28]. Combined with our extensive news article corpus findings, it underlines the significant relevance NSEs hold, highlighting the need for continued research in this domain to ensure that navigation technologies can contribute positively to our society.

## Supporting information

**S1 File. Convenience sample from random search terms.**
(XLSX)

**S2 File. Included and excluded set of keywords the search terms are based on.**
(XLSX)

**S3 File. Search term A analysis with 1000 articles.**
(XLSX)

**S4 File. Search term B analysis with 1000 articles.**
(XLSX)

**S5 File. Search term C analysis with 1000 articles.**
(XLSX)

**S6 File. The coding results of the 90 final news articles.**
(XLSX)

## Author Contributions

**Conceptualization:** Eve Schade, Gian-Luca Savino, Johannes Schöning.

**Data curation:** Eve Schade, Gian-Luca Savino, Yasemin Gunal.

**Formal analysis:** Eve Schade, Gian-Luca Savino, Yasemin Gunal.

**Funding acquisition:** Johannes Schöning.

**Investigation:** Eve Schade, Gian-Luca Savino, Yasemin Gunal.

**Methodology:** Eve Schade, Gian-Luca Savino, Yasemin Gunal, Johannes Schöning.

**Project administration:** Gian-Luca Savino, Johannes Schöning.

**Software:** Eve Schade, Gian-Luca Savino.

**Supervision:** Gian-Luca Savino, Johannes Schöning.

**Validation:** Eve Schade, Gian-Luca Savino, Yasemin Gunal.

**Visualization:** Eve Schade, Gian-Luca Savino, Yasemin Gunal.

**Writing – original draft:** Eve Schade, Gian-Luca Savino, Yasemin Gunal, Johannes Schöning.

**Writing – review & editing:** Eve Schade, Gian-Luca Savino, Johannes Schöning.

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
