## [Decision Letter · Decision Letter 0]

16 Feb 2024

PONE-D-23-40588Traffic jam by GPS: A systematic analysis of the negative social externalities of large-scale navigation technologiesPLOS ONE

Dear Dr. Schade,

Thank you for submitting your manuscript to PLOS ONE. After careful consideration, we feel that it has merit but does not fully meet PLOS ONE’s publication criteria as it currently stands. Therefore, we invite you to submit a revised version of the manuscript that addresses the points raised during the review process.

**In particular, reviewer 2 identifies as one of the stronger contribution to this paper the data collected, which could be informing future research. Accessibility to the data collected is required by PLOS ONE policy and I therefore warmly encourage to upload them to a public repository indicated in the revised version of this manuscript. In addition, as suggested by reviewer 1, the opportunities of integrating the information collected with other datasets can be discussed.**

We look forward to receiving your revised manuscript.

Kind regards,

Riccardo Gallotti

Academic Editor

PLOS ONE

Journal Requirements:

Reviewers' comments:

Reviewer's Responses to Questions

**Comments to the Author**

1. Is the manuscript technically sound, and do the data support the conclusions?

Reviewer #1: Yes

Reviewer #2: Partly

2. Has the statistical analysis been performed appropriately and rigorously? 

Reviewer #1: Yes

Reviewer #2: N/A

3. Have the authors made all data underlying the findings in their manuscript fully available?

Reviewer #1: Yes

Reviewer #2: No

4. Is the manuscript presented in an intelligible fashion and written in standard English?

Reviewer #1: Yes

Reviewer #2: Yes

5. Review Comments to the Author

Reviewer #1: I want to thank the editor for allowing me to review such paper.

Here, the authors formalize a systematic analysis, using a content analysis, of a corpus of news articles that they reported traffic problems and negative social externalities (NSEs) in urban areas caused by the wide usage of navigation technologies. The authors discovered and categorized externalities discovered in the news articles and they found two major categories, traffic-related and safety-related externalities, and they made a statistics of externalites and how often they occur.

The problem is well introduced with the pleasant introduction on the collective problems associated with the wide usage of navigation technologies and their re-routing algorithms, with which they use a user-centred solutions and they move the traffic from highway to small streets in urban areas. The authors also discuss about the solution implemented from government and citizen to reduce externalities and they found that the government solutions had higher success rates than the citizen solutions.

Major remarks

1) Metrics

A systematic analysis of news articles related to the impact of navigation technologies, does not give a measure on how much is the impact of the re-routing taffic to urban areas. How much the traffic, pollution and noise are increased in urban areas in the last decade? Tom Tom has a good API with traffic statistics (https://www.tomtom.com/products/traffic-stats/). The dataset cover several years and countries, it might be interesting to try to see if it is possible to take the cases described in the selected articles and measure the impact of navigation systems using TomTom data. This is only an idea and probably it doesn't work.

2) Datasets

In line 110, why was each search terms run on a set of 1000 articles and not on the entire articles dataset? It could be that in the 2000 items not scanned with one of the search terms there arother relevant articles. This step of the method introduced by the authors is not much clear.

Minor remarks

1) Correlations between negative social expertalities and solution adopted

In the paragraph that stats in row 267, the authors compare the solutions applied to NSEs, if there are enough data, it might be interesting a plot with correlation between the eight categories of externalities and solutions applied (all and only the successful ones). The plot could be more intuitive than a table.

2) In table 1 it could be a good idea to add a column with the category of the externalities (traffic-related and safety-related) because this is not clear if someone read only the table. Also, add in the caption that the percentage refers to the subset of articles related to the one of two externalities categories.

In line 373 there is a word "zeitgeist" that I don't understand what it means. Is it a typo?

Reviewer #2: In this paper, the authors collect and analyze newspaper articles regarding the negative externalities of navigation technologies, providing simple descriptive statistics.

I appreciate the effort you have put into collecting and analyzing newspaper articles on the negative externalities of navigation technologies. I find the dataset intriguing and believe that with some revisions, your paper could better contribute to the scientific discourse, particularly in the rapidly growing field of quantitative assessments of the impact of navigation technologies on the urban environment.

I would like to suggest a shift in the paper's focus from a standard research article to a data paper. A data paper would allow for a more comprehensive exploration of the dataset's potential and open avenues for future research. Additionally, it is crucial to make the dataset publicly accessible by providing a link to ensure broader usage and collaboration within the research community.

Here are specific comments and suggestions for your consideration:

1) Literature Review and Context: The paper lacks a thorough engagement with recent literature on quantitative assessments of externalities related to navigation systems. I recommend discussing the following relevant papers to provide a more comprehensive context:

- https://research.google/pubs/quantifying-the-sustainability-impact-of-google-maps-a-case-study-of-salt-lake-city/

- https://doi.org/10.1145/3557915.3560977

- https://doi.org/10.1142/S0219477524500160

- https://www.nature.com/articles/s43588-023-00469-4

2) Sample Size and Repetition: The sample size of the articles appears small, and there is a risk of duplicate coverage of the same news. I suggest addressing this issue by conducting an analysis to determine how many times the same event is reported across different news sources. This will enhance the robustness of your findings.

3) Clarification on 'Solutions Made by Citizens': The term 'solutions made by citizens' is not sufficiently clear. I recommend providing specific examples or elaborating on what is meant by this phrase to enhance the paper's clarity and understanding.

4) Dataset Accessibility: While the authors have taken steps to make the downloading procedure reproducible, it is equally important to provide a link to the dataset. Given the paper's emphasis on the value of the data, making the dataset publicly available will contribute significantly to the paper's impact and encourage future collaborations.

6. PLOS authors have the option to publish the peer review history of their article (what does this mean?). If published, this will include your full peer review and any attached files.

Reviewer #1: **Yes: **Andrea Guizzo

Reviewer #2: **Yes: **Luca Pappalardo

---

## [Author Response · Author response to Decision Letter 0]

16 Apr 2024

Dear Reviewers,

We would like to express our sincere gratitude for the valuable feedback provided on our submission (PONE-D-23-40588). We truly appreciate your dedicated time and effort in reviewing our work. In response to your insightful comments, we have carefully revised our manuscript, incorporated the suggestions, and addressed the concerns raised. Firstly, we are pleased to note that the reviewers found our paper well-written [R1] and deemed our dataset intriguing [R2]. These affirmations serve as encouraging validation of our efforts. To facilitate a comprehensive understanding of the revisions made, we have highlighted all major changes in blue within the revised manuscript. Additionally, we have structured a detailed response addressing each of the reviewers' comments, directly referencing relevant sections, figures, and tables in the manuscript.

Literature Review and Context: Regarding the literature review and contextualization, we have augmented the introduction to highlight the consequences of negative societal externalities, thereby strengthening the motivation behind our study. These augmentations are further echoed in the discussion section, providing a coherent narrative throughout the manuscript [R2].

Clarification on the number of articles: In response to the request for clarification on the number of articles reviewed per search term, we have provided explicit details in the designated sections of the manuscript, as suggested by R1.

Updated Tables and Results: Moreover, we have meticulously updated the tables with associated categories and references for individual articles, as per the recommendations of R1. To ensure accuracy and reliability, we have thoroughly reviewed and verified all articles, resulting in a more robust presentation of our findings.

Relationship between negative social externalities and solutions: In addressing the relationship between negative social externalities and the proposed solutions, we have conducted an in-depth analysis, which is visually represented in Figure 3 and described in the paper. This graphical representation offers valuable insights into the effectiveness of various solutions, aligning with the suggestions by R1 and enhancing the coherence of our paper.

Throughout the revision process, we have implemented numerous minor edits to enhance clarity, as suggested by both reviewers. These refinements contribute to the overall readability and coherence of the manuscript. Finally, we have updated all supporting information to align with the revisions made in this iteration. Once again, we thank our reviewers for their thoughtful evaluation and constructive criticism. We trust that the changes will further strengthen our work's quality and impact, and we are looking forward to your response. We believe that our paper contributes to understanding the societal implications of technology by identifying issues like congestion, accidents, and pollution caused by navigation technologies. We emphasize the necessity for collaboration among governments, residents, planners, and technologists to mitigate these negative effects. Additionally, we highlight community and governmental responses, such as street closures and speed limit reductions, to address these challenges.

---

## [Decision Letter · Decision Letter 1]

22 Jul 2024

Traffic jam by GPS: A systematic analysis of the negative social externalities of large-scale navigation technologies

PONE-D-23-40588R1

Dear Dr. Schade,

We’re pleased to inform you that your manuscript has been judged scientifically suitable for publication and will be formally accepted for publication once it meets all outstanding technical requirements.

Kind regards,

Xiangjie Kong

Academic Editor

PLOS ONE

Additional Editor Comments (optional):

Reviewers' comments:

Reviewer's Responses to Questions

**Comments to the Author**

1. If the authors have adequately addressed your comments raised in a previous round of review and you feel that this manuscript is now acceptable for publication, you may indicate that here to bypass the “Comments to the Author” section, enter your conflict of interest statement in the “Confidential to Editor” section, and submit your "Accept" recommendation.

Reviewer #1: All comments have been addressed

2. Is the manuscript technically sound, and do the data support the conclusions?

Reviewer #1: Yes

3. Has the statistical analysis been performed appropriately and rigorously? 

Reviewer #1: N/A

4. Have the authors made all data underlying the findings in their manuscript fully available?

Reviewer #1: Yes

5. Is the manuscript presented in an intelligible fashion and written in standard English?

Reviewer #1: Yes

6. Review Comments to the Author

Reviewer #1: I want to thank the editor for allowing me to review such an interesting paper.

Here, the author analyze newspaper in English language about Negative Externalities of navigation technologies and the solution adopted to resolve externalities by citizens and governments.

After first review, the authors enriched the references as required by reviewer 2 and add more informations about the correlations between type of externality and type of solution adopted as required by reviewer 1.

The authors also provides the links to download the articles from NexisUni platform, however, it is not possible to download all articles togheter. It might be interesting to create a repository with all articles

Minor remarks

In rows 97, 106, 127, 132 add in the brackets the reference to Fig.1, e.g (Fig. 1a) or (Fig. 1, step a)

In table 1, add in the caption that the percentage in the third column in the table refers to the entire final corpus.

7. PLOS authors have the option to publish the peer review history of their article (what does this mean?). If published, this will include your full peer review and any attached files.

Reviewer #1: **Yes: **Andrea Guizzo

---

## [Editor Report · Acceptance letter]

26 Jul 2024

PONE-D-23-40588R1 

PLOS ONE

Dear Dr. Schade, 

I'm pleased to inform you that your manuscript has been deemed suitable for publication in PLOS ONE. Congratulations! Your manuscript is now being handed over to our production team.

Kind regards, 

on behalf of

Dr. Xiangjie Kong 

Academic Editor

PLOS ONE